# Resurrection of a global, metagenomically defined gokushovirus

**Paul C Kirchberger\*, Howard Ochman**

Department of Integrative Biology University of Texas, Austin, United States

**Abstract** Gokushoviruses are single-stranded, circular DNA bacteriophages found in metagenomic datasets from diverse ecosystems worldwide, including human gut microbiomes. Despite their ubiquity and abundance, little is known about their biology or host range: Isolates are exceedingly rare, known only from three obligate intracellular bacterial genera. By synthesizing circularized phage genomes from prophages embedded in diverse enteric bacteria, we produced gokushoviruses in an experimentally tractable model system, allowing us to investigate their features and biology. We demonstrate that virions can reliably infect and lysogenize hosts by hijacking a conserved chromosome-dimer resolution system. Sequence motifs required for lysogeny are detectable in other metagenomically defined gokushoviruses; however, we show that even partial motifs enable phages to persist cytoplasmically without leading to collapse of their host culture. This ability to employ multiple, disparate survival strategies is likely key to the long-term persistence and global distribution of *Gokushovirinae*.

## Introduction

Single-stranded circular DNA (ssDNA) phages of the family *Microviridae* are among the most common and rapidly evolving viruses present in the human gut (*Lim et al., 2015*; *Minot et al., 2013*; *Manrique et al., 2017*; *Shkoporov et al., 2019*). Within the *Microviridae*, members of the subfamily *Gokushovirinae* are detected in metagenomic datasets from diverse environments, ranging from methane seeps to stromatolites, termite hindguts, freshwater bogs and the open ocean (*Bryson et al., 2015*; *Desnues et al., 2008*; *Quaiser et al., 2015*; *Tikhe and Husseneder, 2018*; *Tucker et al., 2011*).

Due to their small, circular genomes, full assembly of these phages from metagenomic data is easy (*Creasy et al., 2018*; *Labonté and Suttle, 2013*; *Roux et al., 2012*), and more than a thousand complete metagenome-assembled microvirus genomes have been deposited to NCBI as of beginning of 2020. In contrast, *Microviridae* have been isolated from very few hosts, hardly representative of their diversity as a whole, and the only readily cultivable member of this family is *phiX174*, which is classified to the distantly related subfamily *Bullavirinae* (*Doore and Fane, 2016*; *Krupovic et al., 2016*). While *phiX* and *phiX*-like phages are among the most well-studied groups of viruses, they are rare in nature and occupy a small specialist niche as lytic predators of select strains of *Escherichia coli* (*Michel et al., 2010*). Conversely, the *Gokushovirinae* and several other (though not formally described) subfamilies of *Microviridae* are seemingly abundant in the environment but almost exclusively known from metagenomic datasets (*Creasy et al., 2018*; *Székely and Breitbart, 2016*), although estimates of their actual numbers could be biased due to the methods used to prepare metagenomic samples (*Kim and Bae, 2011*; *Roux et al., 2016*). Despite their apparent prevalence in the environment, the only isolated gokushoviruses are lytic parasites recovered from the host-restricted intracellular bacteria *Spiroplasma*, *Chlamydia* and *Bdellovibrio* (*Brentlinger et al., 2002*; *Garner et al., 2004*; *Ricard et al., 1980*). Given their regular occurrence in metagenomes from diverse habitats, it seems unlikely that *Gokushovirinae* only infect intracellular bacteria, and their lack of recovery from other hosts is puzzling.

**\*For correspondence:**
pkirchberger@utexas.edu

**Competing interests:** The authors declare that no competing interests exist.

Typical gokushoviruses pack their 4000-6000 nt genomes, composed of 3–11 genes, into tailless icosahedral phage capsids (*Roux et al., 2012*). No gokushoviruses encode an integrase, which has led to the assumption that they are lytic phages. However, the presence of prophages belonging to several undescribed groups of microviruses within the genomes of some *Bacteroidetes* and *Alphaproteobacteria* (*Krupovic and Forterre, 2011*; *Zhan and Chen, 2019*; *Zheng et al., 2018*) raises the possibility that some can be integrated through the use of host-proteins, in a similar manner to the XerC/XerD dependent integration of ssDNA *Inoviridae* (*Krupovic and Forterre, 2015*).

All gokushovirus genomes assembled from metagenomic data lack multiple genes that are present in *phiX*-like phages (*Roux et al., 2012*). Markedly absent are: (*i*) a peptidoglycan synthesis inhibitor that leads to host cell lysis (although some phages appear to have horizontally acquired bacterial peptidases) and (*ii*) a major spike protein involved in host cell attachment (*Doore and Fane, 2016*; *Roux et al., 2012*). These proteins represent crucial elements in the infectious cycle of *phiX*, and their absence from other gokushoviruses indicates that these phages might operate quite differently on a molecular level.

Here we detect a large number of gokushovirus prophages integrated into the genomes of enterobacteria, contrasting with their predicted exclusively lytic lifestyle. Through transformation of a synthesized prophage genome into a laboratory strain of *Escherichia coli*, we cultivate a novel gokushovirus capable of lysogenizing enterobacteria. Using this experimental model system, we demonstrate that this phage is capable of passive integration via XerC/XerD mediated recombination of phage encoded *dif*-motifs with their bacterial counterparts. We also show that this capability has evolved independently in multiple lineages of gokushoviruses and demonstrate the existence of an intermediate, pseudolysogenic step between lytic and lysogenic lifestyle, indicating that gokushoviruses can lead a decidedly different existence from that of previously characterized members of the *Gokushovirinae*.

## Results

### A new, diverse group of gokushovirus prophages

Querying fully assembled bacterial genomes with the major capsid protein VP1 of gokushovirus *Chlamydia*-phage 4 (NCBI Gene ID 3703676) returned 95 high-confidence hits (E-value <0.0001) within the Enterobacteriaceae (91 from *Escherichia,* and one each from *Enterobacter*, *Salmonella*, *Citrobacter* and *Kosakonia*; *Supplementary file 1*). Although *Gokushovirinae* were previously known only as lytic phages, inspection of each of the associated genomic regions revealed the presence of integrated prophages 4300–4700 bp in length and having a conserved six-gene arrangement: VP4 (replication initiation protein), VP5 (switch from dsDNA to ssDNA replication protein), VP3 (scaffold protein), VP1 (major capsid protein), VP2 (minor capsid protein) and VP8 (putative DNA-binding protein). Most of the variation in both the genome size and sequence of these prophages is confined to three regions: (*i*) near the C-terminus of VP2, (*ii*) in the non-coding region between VP8 and VP2, and (*iii*) within VP1, whose hypervariability is characteristic of the *Gokushovirinae* (*Chipman et al., 1998*; *Diemer and Stedman, 2016*; *Figure 1A*).

All detected prophages are flanked by *dif*-motifs, which are 28 bp palindromic sites that are known to be the targets of passive integration by phages and other mobile elements (*Blakely et al., 1993*; *Das et al., 2013*). The *dif*-motifs upstream of the insertions are highly conserved and individually differ by, at most, one nucleotide from the canonical *dif*-motif of *Escherichia coli*. These upstream *dif*-motifs consist of a central 6 bp spacer flanked by two 11 bp arms, previously been shown to bind tyrosine recombinases XerC/XerD during chromosome segregation and integration of mobile elements (*Castillo et al., 2017*). In contrast, the *dif*-motifs downstream of detected prophages are more variable, particularly in the spacer region and XerD-binding arms, representing the phage *dif*-motifs integrated along with the phage (*Figure 1A*).

A whole genome phylogeny of gokushovirus prophages shows a number of well-differentiated (bootstrap support 70% or higher) lineages, forming clades A-J that each contains members with >95% average nucleotide identity, and 14 singleton lineages with no close relatives (*Figure 1B*, *Figure 1—figure supplement 1*). Comparing the topology of the phage phylogeny with that of their *E. coli* host strains shows examples of prophage clades found only in specific branches of the *E. coli* phylogeny and clades with wider distributions. For example, whereas prophages belonging to clade

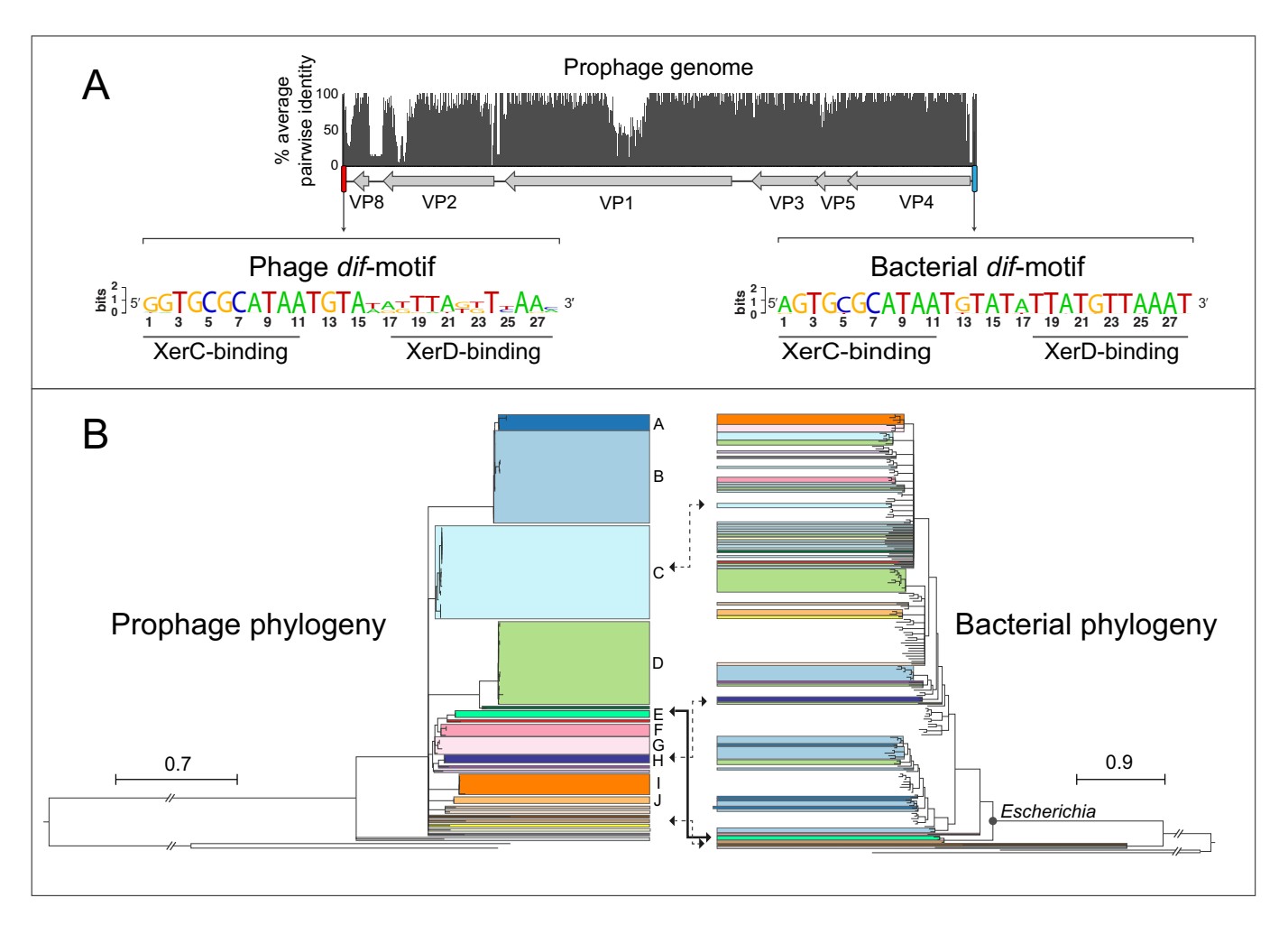

**Figure 1.** Gokushovirus prophages of Enterobacteria. (**A**) Genome organization and average pairwise nucleotide identities of the gokushovirus prophages detected in *Escherichia*. Integrated prophage genomes range from 4316 to 4692 bp in length, with genes indicated by grey arrows, and flanking phage and bacterial *dif*-motifs indicated by red and blue bars. Nucleotide sequence logos of phage and bacterial *dif*-motifs, with corresponding XerC- and XerD-binding sites are shown. (**B**) Phylogeny of gokushovirus prophages and their enterobacterial hosts. Single lineages or clades comprising strains sharing >95% average nucleotide identity are individually colored, with colors in the bacterial phylogeny corresponding to those of their associated prophages. Clades not colored in the bacterial phylogeny correspond to *Escherichia coli* collection reference (ECOR) strains. Clades with bootstrap support values below 70% are collapsed. Arrows denote prophage-host pairs in which the corresponding phage was tested against *E. coli* K-12, with the solid arrow indicating the prophage-host pair whose phage formed plaques in *E. coli* K-12 and subsequently used in experimental analyses. Tree scale bars correspond to nucleotide or amino acid substitutions/site for prophage and host trees, respectively, and ancestral branches with hatch-marks are truncated by the length of two scale bars. Accession numbers and details of prophages and their corresponding hosts are listed in *Supplementary files 1* and *3*.

The online version of this article includes the following figure supplement(s) for figure 1:

**Figure supplement 1.** Sequence similarities among gokushoviral prophages.

A-D are found dispersed across the host phylogeny, prophage clade I is confined to a single *E. coli* clade (possibly as a result of the limited number of detected prophages). In all but one case, each *E. coli* host harbors only a single gokushovirus prophage. Although phage attachment and infection sometimes depend on O-antigenicity, there is no obvious association between the presence of gokushovirus prophages and particular *E. coli* O-serotypes among those strains for which information on O-antigens is available (*Zhou et al., 2020*). For example, four identical prophages belonging to clade C could be detected in *E. coli* strains MOD1-EC5200 (O76:H19), MOD1-EC5181 (O132:H8), NCTC9043 (O43:H2) and MOD1-EC6266 (O124:H21) (*Supplementary file 1*). Most gokushovirus

prophages were detected in diverse *E. coli* strains isolated from various animals, mainly cattle and marmots, but five prophages were detected in isolates from humans, including one from a urinary tract infection (*Supplementary file 1*).

## Prophages of enterobacteria form a distinct gokushovirus clade

Phylogenetic analysis of available gokushovirus genomes based on an alignment of the conserved VP1 and VP4 proteins (*n* = 855; including the enterobacterial prophages discovered in this study, the previously sequenced lytic gokushoviruses from *Chlamydia*, *Spiroplasma* and *Bdellovibrio,* and the metagenomically assembled gokushovirus genomes available from NCBI) returned enterobacterial prophages as a well-supported (97% bootstrap support), monophyletic clade within the *Gokushovirinae* (*Figure 2*). The distinctiveness of this clade, whose members share a conserved (but not unique) gene order and display an average nucleotide identity of >50% (*Figure 1—figure supplement 1* and *Supplementary file 2*), advocates the formation of a new proposed genus within the family *Microviridae* (subfamily *Gokushovirinae*), for which we suggest the name *Enterogokushovirus* on account of a distribution limited to members of the *Enterobactericeae*.

Since a unique feature of enterogokushoviruses among the *Gokushovirinae* appears to be the ability to exist as lysogens, we searched all other gokushovirus MAGs in our dataset for *dif*-like sequence motifs that might be indicative of lysogenic ability. We detected similar motifs in 48 genomes distributed sporadically throughout the gokushovirus phylogeny. Aside from the enterogokushoviruses, there are two larger clades in which multiple genomes contain a *dif*-motif, with the rest of *dif*-bearing MAGs found largely as singletons (*Figure 2*; *Supplementary file 4*). The majority of these putative *dif-motif*s occurs in non-coding regions, and those detected within coding regions were typically at the 5'- or 3'-end of a predicted gene, with a nearby alternative start or stop codon that could preserve the genetic integrity of the phage through integration and excision. The overall dearth of *dif*-like sequences in gokushoviruses sampled from diverse geographic and ecological settings highlights the distinctiveness of *Enterogokushovirus* genomes and lifestyle.

We further attempted to determine the prevalence of enterobacterial gokushoviruses by interrogating 1839 samples from eight metagenomic studies of human and cattle gut microbiomes for the presence of closely related prophages. From these data, we were able to fully assemble only two integrated prophages corresponding to *E. coli* gokushoviruses, one from the fecal metagenome of an Austrian adult (0,05% of all reads in ERR688616) and the other from a Danish infant (0,02% of all reads in ERR525761) (see also *Supplementary file 4*).

## Reconstituting viable phage from integrated prophages

The integrity of prophage structure in all enterobacteria and the lack of premature stop codons suggested that these sequences represent intact, functional insertions into bacterial hosts. To confirm the functionality of *Escherichia* gokushovirus prophages, characterize their biology and provide a type strain, we attempted to construct phages from genomic DNA of *Escherichia* strains MOD1-EC2703, MOD1-EC5150, MOD1-EC6098 and MOD1-EC6163, selected to represent the diversity of gokushovirus prophages and hosts (*Figure 1B*).

Sequences corresponding to prophages from these four *Escherichia* strains were amplified, circularized and transformed into *E. coli* DH5α (*Figure 3A*). Supernatants from the transformed DH5α culture were used in agar-overlay assays with *E. coli* K12 BW25113 hosts, resulting in plaques for only one of four reconstructed phage genomes (EC6098, derived from *E. marmotae* strain MOD1-EC6098 and belonging to prophage clade E (*Figure 3B*, *Figure 3—figure supplement 1A*). While both DH5α and BW25113 possess other prophages which could produce false positives when screening for gokushovirus production (*Chen et al., 2018*; *Wang et al., 2010*), DH5α alone did not produce plaques on BW25113 and vice versa (*Figure 3—figure supplement 1*). To corroborate the synthesis of EC6098 phages (and the inability to produce phages from the other three sequences), we additionally grew live cultures of MOD1-EC2703, MOD1-EC5150, MOD1-EC6098 and MOD1-EC6163 and via PCR detected the presence of circularized gokushovirus genomes in all four, but only MOD1-EC6098 produced plaques resembling those derived from synthetic EC6098 (*Figure 3—figure supplement 1B*). As only EC6098 formed plaques on K12 strains, we used this phage for further experimental characterization.

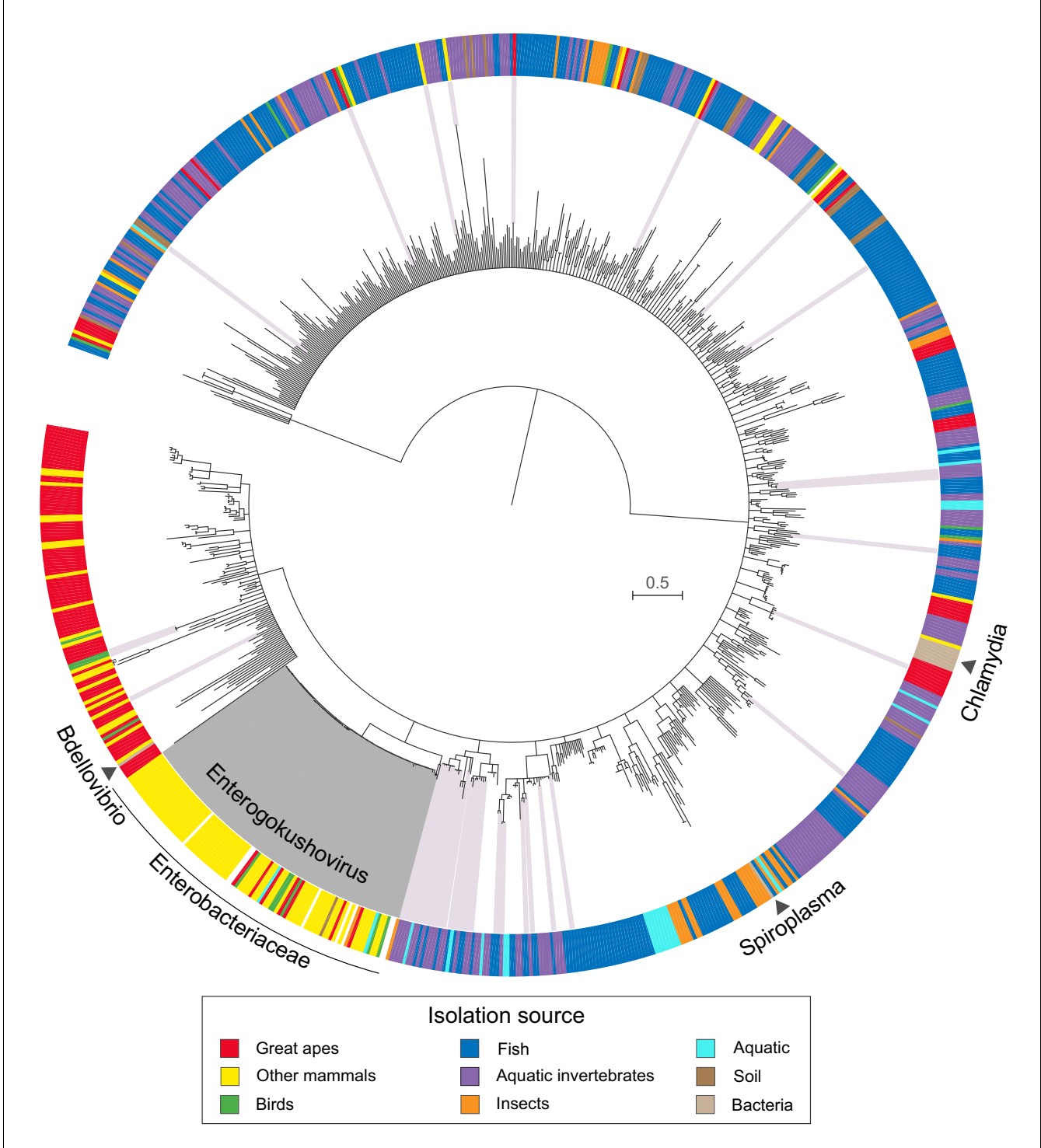

**Figure 2.** Phylogeny and sources of *Gokushovirinae*. Maximum likelihood tree built from concatenated alignments of VP1 and VP4 protein sequences of 855 gokushovirus genomes. Tree is midpoint rooted, and branch support estimated with 100 bootstrap replicates. Branches with bootstrap support values below 70% are collapsed. Clades highlighted in grey indicate *Enterogokushovirus* prophages with recognizable *dif*-motifs; those highlighted in pink possess *dif*-motifs identified through an iterative HMM search. Outer ring indicates isolation source, with black triangles denoting the phylogenetic positions of officially described gokushoviruses. Scale bar corresponds to amino acid substitutions per site. Sample accession numbers are listed in *Supplementary file 4*.

Electron microscopic observation of pure EC6098 lysates recovered at a buoyant density of around 1.3 g/cm$^{-3}$ in CsCl gradients (the range expected for members of the Microviridae, *Thurber et al., 2009*), revealed tailless icosahedral virions, 25–30 μm in size and displaying protrusions previously observed in other *Gokushovirinae* (*Chipman et al., 1998*; *Diemer and Stedman, 2016*; *Figure 3D,E*). Because *Gokushovirinae* have only been recovered as lytic particles from intracellular bacteria, this represents the first isolation of a gokushovirus able to infect free-living bacteria as well as being able to integrate as a lysogen into bacterial genomes.

## Mechanisms of enterobacterial gokushovirus integration into host genomes

We next investigated the process by which gokushoviruses integrate into the bacterial host chromosome (*Figure 4A*). The presence of circularized phage genomes could readily be detected from surviving colonies in confluent plaques derived from agar-overlay assays (*Figure 4B*). Using primers that flank both sides of the *dif*-motif in host strain BW25113, we recovered products that were enlarged by the length of the phage relative to colonies lacking the prophage (*Figure 4A,C*). Sequencing confirmed that, in accordance with the integration site of detected prophages, phage EC6098 integrates downstream of the BW25113 *dif*-motif, which remains unchanged (corresponding to position 3,046,436–3,046,463 in the closed genome with accession number CP009273).

Because none of the gokushovirus prophages encodes an integrase, we predicted that host factors XerC and XerD might be responsible for prophage integration, similar to what has been hypothesized for microvirus-prophages in *Bacteroidetes* (*Krupovic and Forterre, 2011*). Neither ΔxerC nor ΔxerD mutants of *E. coli* host strain BW25113 resulted in integration, but the ΔxerC mutant was restored by complementation with a plasmid expressing the intact version of *xerC*. Similarly, phages with incomplete (*i.e.*, lacking either their XerC or XerD binding site within the *dif*-motif, termed here ΔdifC and ΔdifD) or no *dif*-motifs (ΔdifCD) successfully infected hosts (as evidenced by the presence of circularized phage genomes in host colonies that had survived infection), but failed to integrate into host genomes, demonstrating the need for cooperative XerC/XerD binding for successful lysogeny (*Table 1*, *Figure 4B,C*). However, the retention of *dif-moti*fs after integration indicates that this process is reversible: As evident in *Figure 4C*, there is a smaller fragment, in addition to that indicating prophage insertion, that corresponds to those cells in the same colony that do not harbor the integrated prophage. These cells persisted even after multiple rounds of re-streaking, and the median ratio of lysogens to non-lysogenic cells derived from clonal colonies approaches 4:1 (*Figure 4D*). Even in pure cultures, phages are thus continuously being excised and reintegrated, presumably as a result of XerC/XerD activity.

## Integration into host genomes facilitates but is not required for long-term persistence

The removal of host factors *xerC* or *xerD*, the presence an incomplete phage *dif-motif*, or the lack of a *dif-motif* (as observed in the majority of *Gokushovirinae*) all prevent integration of phages into host genomes (*Table 1*, *Figure 4C*). However, almost all bacterial colonies that survived infections, regardless of whether they were lysogenized or not, were found to contain circular phage DNA (*Figure 4B*), and the presence of circular phage DNA alone, regardless of integration into the host genome, conferred superinfection immunity (*Figure 5A*). In addition, cultures in which phage DNA could only be detected in circularized form were capable of producing infectious particles at levels significantly higher than those of cultures containing integrated prophages (*Figure 5B*, Wilcoxon rank-sum test p<0.01).

The absence of alternative integration sites, as verified by inverse PCR, suggested that gokushoviruses might be able to persist in the host cytoplasm without integration into the host genome. To determine whether this persistence is a transient phenomenon that eventually leads to either the loss of phages or the collapse of bacterial cultures, we performed serial transfers (1:1000 dilution) with a variety of host-phage combinations over the course of a month. We enumerated plaque formation from culture supernatant on host strain BW25113ΔfhuA to avoid false positives from widespread contaminating dsDNA phages such as phi80 and SW-1, which unlike EC6098 require outer membrane receptor FhuA for successful infection (*Rotman et al., 2012*; *Song et al., 2019*). As expected, all lysogenic cultures (*i.e.*, wild type cultures with integrated gokushovirus prophages)

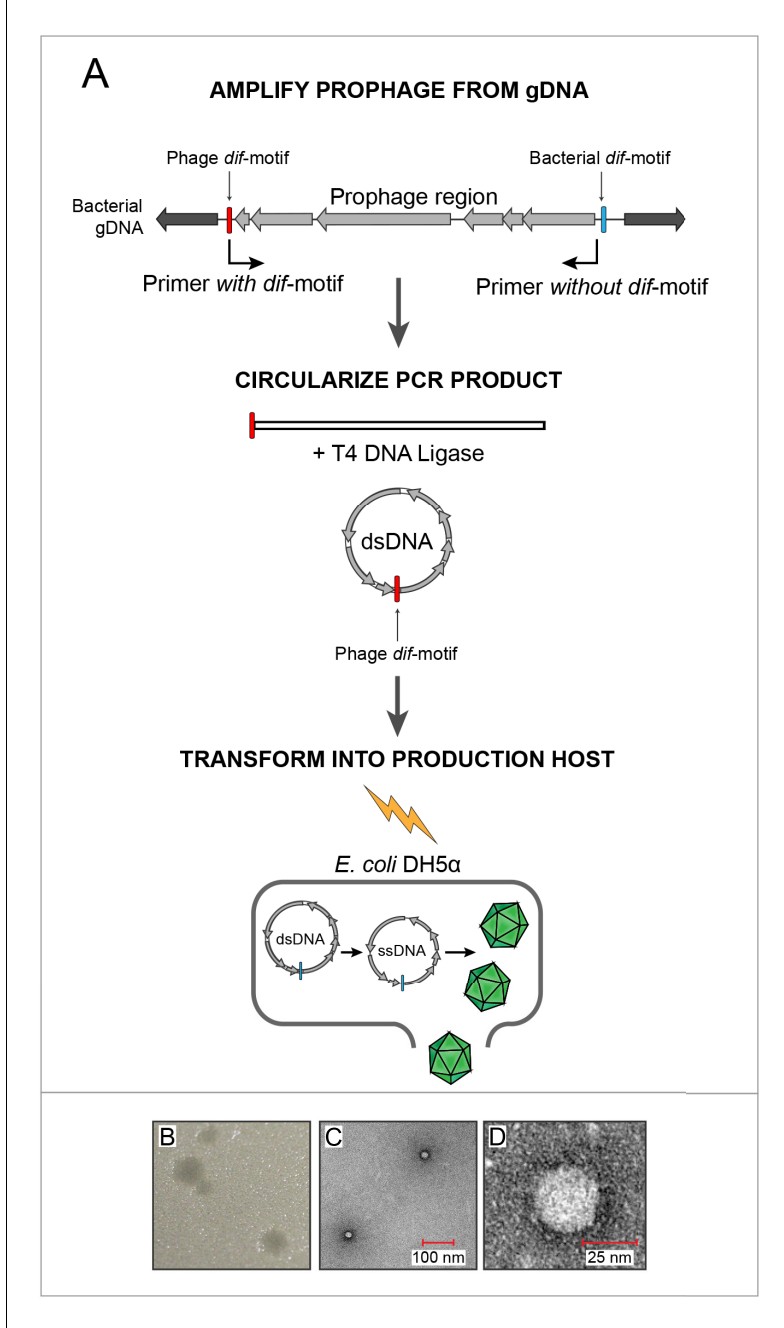

**Figure 3.** In vitro assembly and revival of enterobacterial gokushoviruses. (**A**) Scheme used to produce viable phage from prophage inserts. The prophage region, with genes colored in grey, is amplified from the bacterial genome (black) using primers that incorporate the phage *dif*-motif but exclude the bacterial *dif*-motif (indicated by bent arrows). Circularization of the amplification product results in a molecule corresponding to the replicative dsDNA form of the phage. Transformation of this circular molecule into electrocompetent *E. coli* DH5α cells leads to expression of phage proteins, replication and packaging of ssDNA into infective virions. (**B**) Plaques formed by constructed bacteriophage EC6098 after infecting *E. coli* BW25113. (**C**) TEM images of bacteriophage EC6098 viewed at 175,000x magnification. (**D**) TEM images of bacteriophage EC6098 viewed at 300,000x magnification. The online version of this article includes the following figure supplement(s) for figure 3:

**Figure supplement 1.** Synthesis of gokushoviruses in *Escherichia coli* DH5α and natural host strains.

produced phage particles throughout the course of the experiments. However, a third of all cultures where gokushoviruses were initially present only in the cytoplasm (due to deletions in the host integration machinery or phage integration site) retained phage production over the course of a month despite the lack of integration (*Figure 5C*). Cumulatively, these results indicate that gokushoviruses do not exist solely as lytic particles, as previously believed, but can also exist as (pseudo)lysogens, in a carrier state that continuously produces phage particles without integrating into the genome.

## Discussion

Through the analysis of whole genome sequences and metagenomic databases, we defined a unique genus of *Gokushovirinae* prophages and subsequently synthesized a viable gokushovirus capable of infecting and integrating into the genomes of enteric bacteria. Since *Gokushovirinae* were previously known as exclusively lytic predators of a few intracellular bacteria (*King et al., 2011*), the new proposed genus, *Enterogokushovirus*, offers new insights into our understanding of this ecologically widespread group of phages. First, by confirming the existence of *Gokushovirinae* prophages in free-living bacteria, we resolved a seeming paradox in which a diverse and widespread lineage within the *Microviridae* appeared to be confined to rare pathogenic and parasitic bacteria, such as *Chlamydia* (*Wang et al., 2019*). Second, by demonstrating the integration of a gokushovirus into the *E. coli* genome, we show that these phages employ survival strategies beyond the lytic infection of hosts.

Experimental characterization of the enterogokushovirus EC6098 isolated from an environmental strain of *E. marmotae* showed that these phages possess a *dif*-like recognition motif that, together with the host-encoded recombinases XerC/XerD, is required for lysogeny. In this manner, enterogokushoviruses appropriate a highly conserved bacterial chromosome concatemer-resolution system that enables their integration into host genomes via homologous recombination (*Castillo et al., 2017*). The exact mechanism of this integration process still remains to be elucidated: some phages and mobile elements are known to have different requirements for the bacterial XerC and/or XerD proteins (*Midonet and Barre, 2015*), which is perhaps the reason why we could only successfully reconstitute *xerC* but not *xerD* knockouts. We also show that phages with only partial DNA-binding motifs exist in a condition in which circularized phage genomes are present in the cytoplasm and virions are continuously released from the host. While it should be noted that this could be a laboratory-induced phenomenon, this strategy is similar to the pseudolysogenic state observed in crAssphage (*Shkoporov et al., 2018*) and other bacteriophages (*Siringan et al., 2014*), and helps explain the persistence of microviruses in the human gut (*Minot et al., 2013*; *Shkoporov et al., 2019*).

Despite an exhaustive sampling of gokushoviral diversity from metagenomic datasets, the occurrence of *dif*-positive gokushoviruses is rare outside of the enterogokushoviruses. However, the

**Table 1.** Percentage of lysogenic colonies after phage infection*.

| Strain | Phage | Plasmid | % Lysogens |
|---|---|---|---|
| BW25113 | EC6098 | - | 17.71 |
| BW25113Δ*xerC* | EC6098 | - | 0 |
| BW25113Δ*xerC* | EC6098 | pJN105::*xerC* (induced[†]) | 14.58 |
| BW25113Δ*xerC* | EC6098 | pJN105::*xerC* (uninduced) | 0 |
| BW25113Δ*xerD* | EC6098 | - | 0 |
| BW25113Δ*xerD* | EC6098 | pJN105::*xerD* (induced[†]) | 0 |
| BW25113Δ*xerD* | EC6098 | pJN105::*xerD* (uninduced) | 0 |
| BW25113 | EC6098Δ*dif*C | - | 0 |
| BW25113 | EC6098Δ*dif*D | - | 0 |
| BW25113 | EC6098Δ*difCD* | - | 0 |

*Assessed from screening 96 colonies for each strain.
[†]Expression induced by addition of 0.1% arabinose.

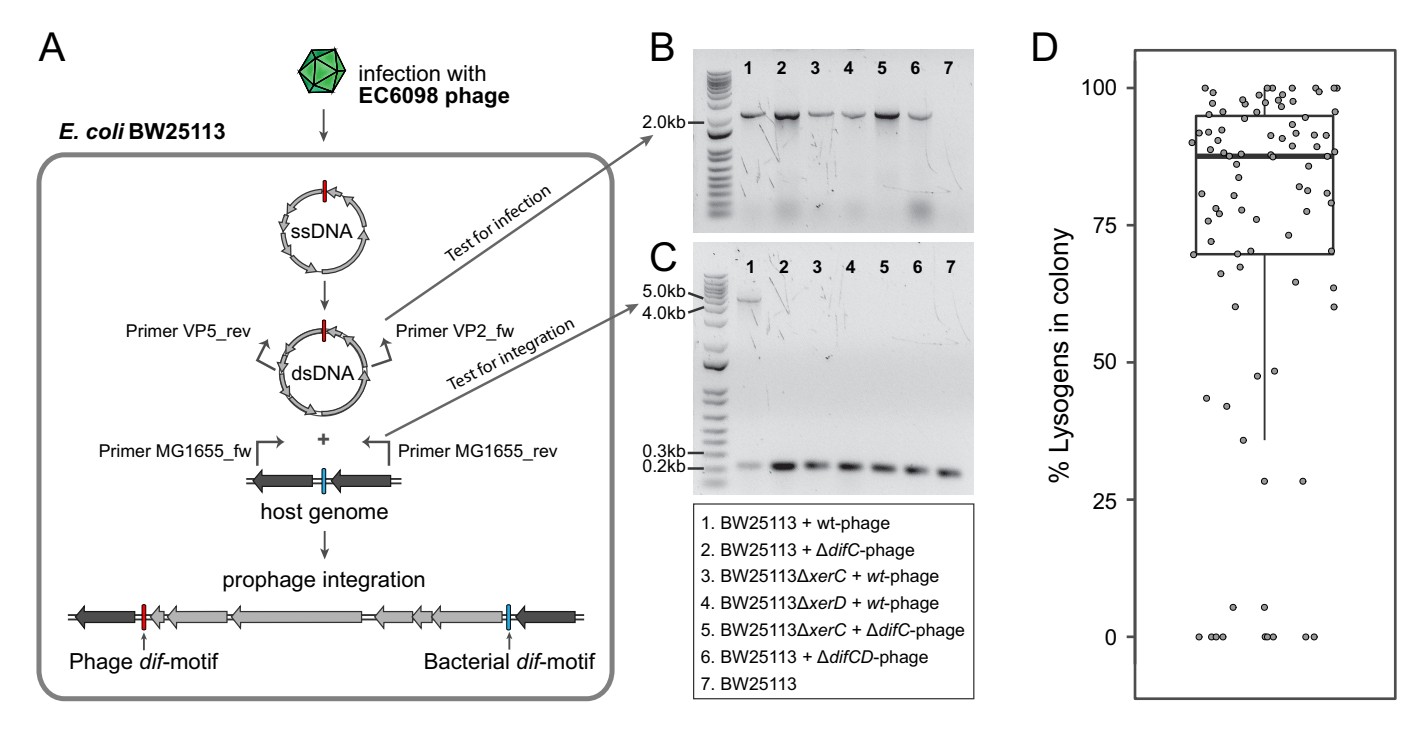

**Figure 4.** Integration of gokushoviruses into *E. coli* host genome. (**A**) Schematic representation of phage integration process, and detection of circularized EC6098 phage genome and integrated prophages in *E. coli* BW25113. Upon infection, phage EC6098 releases ssDNA, with *dif*-motif denoted in red, leading to formation of dsDNA replicative genomes that integrate downstream of the bacterial *dif*-motif (blue). Primers VP2_rev and VP5_fw (indicated by bent arrows on EC6098 genome) anneal to genes flanking the phage *dif*-motif and amplify a ~ 2.1 kb product from closed circular phage genomes (corresponding to bands in panel B). Primers MG1655dif_fw and MG1655dif_rev (indicated by bent arrows on host genome) anneal to sites flanking the bacterial *dif*-motif and amplify either a 210 bp region of bacterial DNA when there is no phage integration or a ~ 5 kb region denoting the presence of an integrated prophage (corresponding to bands in panel C). (**B**) Detection of fragments indicating the presence of circularized phage. Numbered lanes correspond to samples listed in the box below the gel photographs. (**C**) Detection of fragments indicating the presence or absence of integrated phage from lysogenic colonies after infection of BW25113 strains with wild type or mutant phage. Numbered lanes are the same as in panel B, and correspond to samples listed in the box below the gel photograph. (**D**) Proportion of cells with integrated prophages in clonal lysogenic colonies. Box-and-whiskers plot shows median, 25th and 75th percentiles, and 1.5 inter-quartile range as well as individual datapoints for 87 independently sampled clonal colonies.

sporadic distribution of *dif*-motifs among other gokushoviruses indicates that the ability to lysogenize bacterial hosts has been independently gained and lost multiple times during the evolution of this taxon. For example, the exclusively lytic gokushoviruses of *Chlamydia* (*Śliwa-Dominiak et al., 2013*) might once have been able to integrate into their hosts' genomes since extant *Chlamydia* genomes contain coding sequences similar to the gokushoviral replication initiation and minor capsid proteins (*Read et al., 2000*; *Rosenwald et al., 2014*).

We observed that lysogenic cultures produce a considerably lower number of phage particles than non-lysogenic cultures but are less prone to the loss of phage, indicating that the lytic and lysogenic lifecycles confer different tradeoffs. It has long been suggested that lysogeny is a 'safe' strategy for phages in situations where hosts are scarce, with vertical inheritance preventing the dilution of phage particles in the absence of hosts (*e.g.*, *Berngruber et al., 2010*). Meanwhile, lysis (and its more abundant production of phage particles) is advantageous when host availability is high.

Lytic microviruses (*i.e.*, those lacking *dif*-motifs) in the human gut have been predicted to infect Bacteroidetes and other dominant members of that community (*Shkoporov et al., 2019*). However, the enterobacterial hosts of the gokushovirus prophages described in this study generally do not reach very high abundances in the gut microbiome, usually constituting less than 0.1% of the bacterial population (*Human Microbiome Project Consortium, 2012*), prompting the evolution of lysogeny. Perhaps due to the high mutation rate of ssDNA phages, typically two orders of magnitude higher than in dsDNA phages (*Sanjuán et al., 2010*), and an estimated substitution rate of $10^{-5}$/

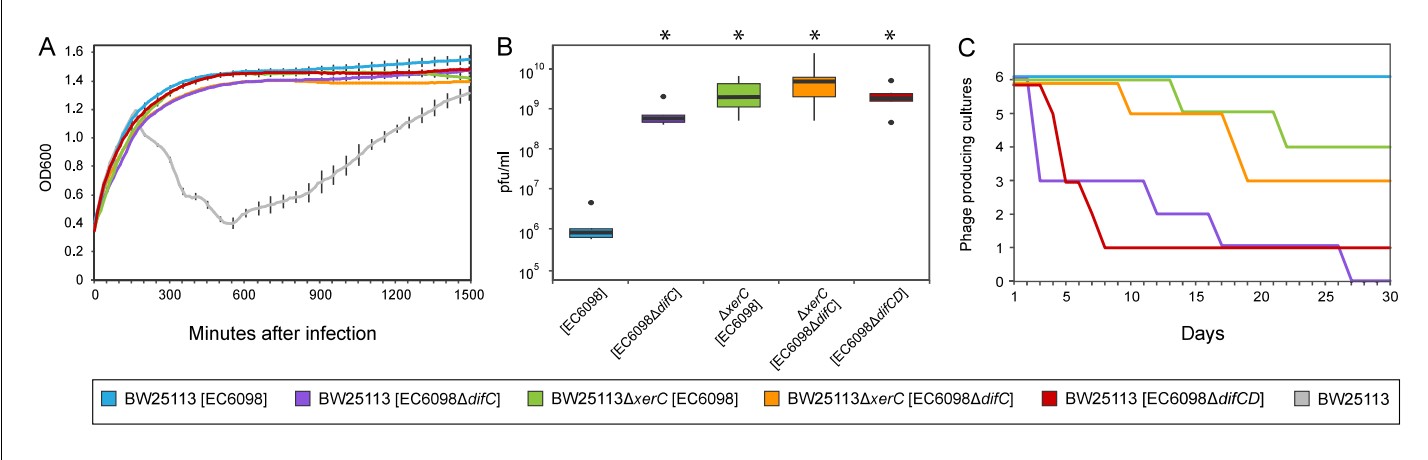

**Figure 5.** Quantifying the effects of gokushovirus carriage. (**A**) Superinfection immunity offered by integrated or non-integrated phages. Unimpeded growth in the presence of infecting EC6098 phage is shown for host cultures carrying lysogenic or lysogeny-deficient EC6098. Lysis of host by EC6098 in the absence of superinfection immunity is shown in grey for comparison. Averages and standard deviations are based on three replicates. (**B**) Phage production by lysogenic and non-lysogenic hosts. Plaque-forming units in culture supernatant were determined from six independent cultures of BW25113 or BW25113ΔxerC, each grown overnight at 37°C from an initial concentration of $OD_{600}$ = 0.7. Box-and-whiskers plots show median, 25th and 75th percentiles (upper and lower hinges) and 1.5 inter-quartile range (whiskers). Outliers are shown as individual dots. P-values of <0.01 (Wilcoxon Rank-Sum test) in comparison to BW25113[EC6098] are indicated with asterisks. (**C**) Continuous phage production over the course of a month. Formation of plaques from supernatants of six replicate cultures of hosts carrying integrated or non-integrated phages is checked daily before 1:1000 dilution and growth for 18–24 hr at 37°C. In all cases, BW25113ΔfhuA served as the host strain for agar overlay assays.

nucleotides/day in the human gut (*Minot et al., 2013*), the de novo evolution of the XerCD-binding motifs appears to be common, with similar systems existing in other families of phage and mobile-elements (*Das et al., 2013*).

Given the current depths of microbiome sampling and sequencing, it is surprising that enterogo-kushoviruses have not previously been identified in human metagenomes. Even among the more than 100,000 sequenced enterobacterial genomes that are currently available, we detected fewer than 100 gokushovirus prophages, and no gokushovirus prophages were detected in the sequenced genomes of other bacteria. This rarity of gokushoviruses existing as prophages might result from the process by which they integrate into genomes: although possession of *dif*-motifs provides a simple, passive integration system, they also facilitate prophage excision (and as a result, continuous pro-duction of phage particles) in the absence of external stressors. In fact, XerCD-mediated removal of genetic elements has attained application in molecular biology (*Bloor and Cranenburgh, 2006*). To prevent premature XerCD-mediated excision, the *Vibrio cholerae* phage CTX secures itself as a pro-phage by destroying the *dif* site upon insertion (*Val et al., 2005*), whereas *Vibrio* prophages that retain their *dif*-motifs are only rarely detected (*Das et al., 2011*). The abundance of gokushoviruses in the environment, as apparent from their regularity in metagenomic datasets, suggests that they are only transient residents of bacterial chromosomes and usually occur in pseudolysogenic or lytic states in the wild.

The discovery of this group of phages offers several new directions for the study of the *Microviri-dae*. First, the presence of numerous and diverse gokushovirus prophages in a wide variety of *E. coli* strains now makes it possible to elucidate aspects of gokushovirus biology in a comparative evolu-tionary framework. Furthermore, demonstrating that the host ranges of the *Gokushovirinae* extends beyond intracellular bacteria increases the likelihood that additional members of this prevalent group of phages will be isolated from appropriate hosts. Additionally, the amplification of entire ssDNA phage genomes and subsequent transformation into appropriate hosts, as conducted in this study and previously with de novo synthesized *phiX* (*Smith et al., 2003*) can aid in the description of other *Microviridae* subfamilies. A promising target for this would be the *Alpavirinae*, which have been detected as prophages of *Bacteroidetes* but so far remain without isolates and denied official recognition (*Roux et al., 2012*). In conclusion, the value of isolating enterogokushoviruses goes

beyond their abundance in nature and provides a genetically manipulable model system that will further the understanding of this group as a whole (*Shkoporov and Hill, 2019*).

## Materials and methods

### Detection and phylogeny of gokushovirus prophages and their hosts

Using blastp, we queried the NCBI nr database (April 2019) with the *Chlamydia* Phage four major capsid protein VP1 (NCBI Gene ID 3703676) and downloaded the complete genomes of 95 strains within the *Enterobacteriaceae* containing sequences returning E-values less than 0.0001 (*Supplementary file 1*). Chromosome contigs containing the VP1 gene were visually inspected in Geneious R9 (www.geneious.com) for the presence of prophage insertion boundaries by searching for identical 17 bp sequences within 5 kb regions upstream and downstream of the VP1 gene. Prophage genes were annotated with GLIMMER3 (*Delcher et al., 2007*) using default settings, specifying a minimum gene length of 110 bp and a maximum overlap of 50 bp. Initial alignments of prophage regions (ranging in size from 4047 to 4692 bp) were made with ClustalO 1.2.4 using standard settings (*Sievers et al., 2011*), and were refined manually to accommodate hypervariable regions and the phage insertion sites at the 3' and 5' ends of the alignment. Average pairwise nucleotide identity at each position in the alignment was calculated and visualized using Geneious R9. Maximum likelihood phylogenetic trees of enterobacterial prophages were generated with RAxML 8.0.26 (*Stamatakis, 2014*) using the GTR+GAMMA substitution model and 100 fast-bootstrap replicates and visualized with FigTree 1.4.3 (http://tree.bio.ed.ac.uk/software/figtree/).

To evaluate the distribution of prophage hosts within the broad diversity of *E. coli* at large, we produced core genome alignments of prophage hosts and representative genomes from the *Escherichia coli* reference (ECOR) collection (*Ochman and Selander, 1984*; *Supplementary files 1* and *3*) based on protein families satisfying a 30% amino-acid identity cutoff (USEARCH 11, *Edgar, 2010*), which were aligned in MUSCLE 3.8.31 (*Edgar, 2004*) as implemented in the BPGA 1.3 pipeline (*Chaudhari et al., 2016*). The maximum likelihood phylogenetic tree of core genome alignments was built with IQTree 1.6.2 (*Nguyen et al., 2015*), using the JTT substitution model (*Jones et al., 1992*) and 100 bootstrap replicates.

### Recovering prophages from metagenomes

To assemble prophages from metagenomic datasets, we downloaded SRA files from BioProjects PRJEB29491 (viral human, *Moreno-Gallego et al., 2019*), PRJNA362629 (cellular bovine, unpublished), PRJNA290380 (cellular human, *Kostic et al., 2015*), PRJNA352475 (cellular human, *Ferretti et al., 2018*), PRJEB6456 (cellular human, *Bäckhed et al., 2015*), PRJNA385126 (viral human, *Stockdale et al., 2018*),PRJEB7774 (cellular human, *Feng et al., 2015*), and PRJNA545408 (human viral, *Shkoporov et al., 2019*). We performed initial trimming and quality filtering with BBDuk (*Bushnell, 2014a*) with options ktrim = r k = 23 mink = 11 hdist = 1 tbe tbo. Reads having a minimum nucleotide sequence identity of 50% to sequences of enterobacterial prophages, as determined by BBMap (*Bushnell, 2014b*), were assembled into contigs using MEGAHIT 1.1.3 (*Li et al., 2015*) implemented with default settings, and those contigs > 1000 bp were retained.

### Phylogenetic analysis of gokushovirinae

We downloaded a total of 1284 metagenome-assembled genomes (MAGs) of microviruses (*Supplementary file 4*), which were then reannotated in GLIMMER3 (*Delcher et al., 2007*) using default settings, with a minimum gene length of 110 bp and a maximum overlap of 50 bp. We recovered homologues to the conserved major capsid protein VP1 and replication initiation protein VP4 in the set of metagenome-assembled microviruses using PSI-BLAST searches and querying with VP1 and VP4 proteins from detected enterobacterial gokushoviruses, gokushovirus genomes of *Chlamydia*, *Spiroplasma* and *Bdellovibrio*, and *Bullavirinae* phage *phiX*174. After individual protein alignments using Clustal Omega 1.2.4 (standard settings), we concatenated the VP1 and VP4 alignments, and removed all sites with >10% gaps to decrease the amount of spuriously aligned sites using Geneious R9, for an alignment of 485 aa in length. The initial phylogenetic tree of all microviruses was built with IQTree 1.6.2 using the LG+F+R10 substitution model as determined by ModelFinder (*Kalyaanamoorthy et al., 2017*), and branch support was tested using 1000 ultra-fast bootstrap

replicates (*Hoang et al., 2018*) and 1000 SH-aLRT tests. Collapsing all branches with <95% bootstrap support and <80% SH-aLRT support yielded a single, well-supported clade containing all known *Gokushovirinae*, and all subsequent alignments and phylogenetic trees were refined by including only those genomes represented in this clade, with branch support assessed with 100 bootstrap replicates. To complement already existing annotations of gokushovirus genomes for the purpose of comparing gene order in various branches of the phylogeny, iterative tblastx searches of individual enterogokushovirus EC6098 genes agains members of select branches were conducted. Hits with e-value <0.001 (lower than initial searches used to identify prophages to account for the larger phylogenetic distance or analyzed genomes) were considered homologs to enterogokushoviral genes.

## Identification of *dif*-motifs in gokushovirus MAGs

To search for *dif*-motifs in enterobacterial prophages, we first performed an alignment of all enterobacterial prophage *dif-motif* sequences in the curated set of bacterial *dif*-motifs from *Kono et al. (2011)* using Clustal Omega 1.2.4. We used the resulting alignment to build a Hidden-Markov-Model using hmmer 3.2.1 (*Wheeler and Eddy, 2013*) and performed an iterative search for *dif*-like motifs in all gokushovirus-like MAGs. Due to the variation in phage and bacterial *dif*-motifs, the variation in these motifs among bacteria, and the short length of the target sites, only confirmed *dif*-motifs of enterobacterial prophages reached an E-value cutoff >0.005. A large number of hits fell below of this threshold and were treated as potential *dif*-motifs if they possessed at least 15 bp identical to confirmed *dif*-motifs and occurred in the short non-coding regions of MAGs. Hits within coding regions were removed as likely representing false positives (as integration would interrupt coding sequences), with the exception of those within the N-terminus of VP4 (as occasionally observed in *Escherichia* gokushovirus prophages).

## Resurrection and modification of prophages

DNA fragments representing gokushovirus prophages were amplified from *E. coli and E. marmotae* strains MOD1-EC2703, MOD1-EC5150, MOD1-EC6098 and MOD1-EC6163 (*Supplementary file 1*) with Phusion polymerase (NEB) from 10 ng of genomic DNA using primer pairs listed in *Supplementary file 5* and under the following PCR conditions: 98°C for 3 min; 30 cycles of 98°C for 15 s, 50°C for 15 s, 72°C 2:30 min; followed by 72°C 10 min. Amplified fragments of ~4.5 kb corresponding to gokushovirus prophages were purified from agarose gels using the Monarch DNA Gel Extraction Kit (NEB) and eluted in 20 µl ddH$_2$O. Blunt ends of the purified linear fragment were phosphorylated with T4 Polynucleotide Kinase (ThermoFisher) followed by overnight treatment with T4 DNA Ligase (NEB) to form circular genomes. Ligation mixtures were heat-inactivated, desalted, transformed into *E. coli* DH5α and incubated for 1 hr in 1 ml SOC medium at 37°C. After this recovery period, cultures were grown overnight in 5 ml of LB medium at 37°C with mild shaking (200 rpm). Viable bacteriophages were harvested by centrifuging the culture for 5 min at 5000 g to pellet bacterial cells and then by filtering the supernatant through 0.45 µm syringe filters. The presence and identity of phages were confirmed through standard spot assays (see below) and Sanger sequencing (see *Supplementary file 5*).

## Plasmid construction and complementation of knockout mutants

To construct complementation plasmids, we first amplified the *xerC* and *xerD* genes from *Escherichia coli* BW25113 with primers XerC_fw_EcoRI and XerC_rev _SacI or XerD_fw_EcoRI and XerD_rev_SacI (*Supplementary file 5*) under the conditions listed above. PCR products were purified using the Monarch DNA Gel Extraction Kit (NEB) and eluted in 20 µl ddH$_2$O. PCR products and expression plasmid pJN105 (*Newman and Fuqua, 1999*) were digested with *Eco*RI and *Sac*I (NEB) for 37°C for 1 hr, followed by heat inactivation for 10 min at 80°C and overnight ligation at a 1:3 vector-to-insert ratio using T4 DNA Ligase (NEB) at 4°C. One microliter of ligation mixtures were transformed into electrocompetent BW25113Δ*xerC or* Δ*xerD* mutants, and transformants were selected for growth on LB agar plates supplemented with 10 µg/ml gentamycin.

## Phage and bacterial culture

Environmental *Escherichia* strains MOD1-EC2703, MOD1-EC5150, MOD1-EC6098 and MOD1-EC6163, *Escherichia coli* K12 derivates DH5α and BW25113, and KEIO collection strains BW25113Δ*xerC* (KEIO Strain JW3784-1), BW25133Δ*xerD* (KEIO Strain JW2862-1) BW25113Δ*fhuA* (KEIO strain JW0146-2) were grown at 37°C in LB liquid media (supplemented with of 50 µg/ml kanamycin for the KEIO knockout strains). Expression of *xerC* or *xerD* genes in BW25113Δ*xerC* and BW25133Δ*xerD* containing pJN::xerC or pJN::xerD was induced by addition of 0.1% arabinose.

To prepare agar-overlays, cells from 100 µl of overnight culture were pelleted, resuspended in PBS, combined with 100 µl of phage and incubated at room temperature for 5 min prior to addition 3 ml of 0.6% LB-agarose and plating onto LB agar. To increase the phage concentrations, we harvested phage lysates from plates exhibiting confluent lysis after overnight growth at 37°C. Phage titers were determined by spotting dilutions of lysates onto agar-overlay plates with 100 µl of overnight cultures of host strains and incubating plates overnight at 37°C. Liquid-infection assays were performed in 96-well plates by adding 2 µl of phage lysate (~$10^8$ pfu/ml) to 200 µl of overnight culture diluted with LB to $OD_{600}$ = 0.4 and measuring growth and lysis at 37°C with 200 rpm shaking at 15 min intervals on a Tecan Spark 10M plate reader.

Serial transfers experiments were performed by inoculating lysogenic colonies (as identified by PCR, below) in LB, diluting overnight cultures to $OD_{600}$ = 0.7, and then transferring 2 µl of the diluted culture into 2 ml of LB. After 18–24 hr incubation at 37°C with shaking, 2 µl of culture was transferred to 2 ml of fresh LB. This process was repeated for 28 days, and each day, phages were titered as described above.

## Detection of circularized phages and prophages

The presence of lysogens and circularized phage was determined by PCR assays of liquid overnight cultures from surviving colonies in agar-overlay assays showing confluent lysis, using primers MG1655_fw and MG1655_rev, which flank the bacterial *dif*-motif, and VP2_fw and VP5_rev, which anneal up- and downstream the phage *dif*-motif in circularized phage genomes (*Supplementary file 5*). Ratios of lysogenic to non-lysogenic cells in individual cultures were measured by re-streaking single lysogenic colonies three times, selecting and resuspending a single resulting colony in LB, and re-plating it onto LB-agar plates. Colonies were grown in liquid culture in a 96-well plate overnight and assayed by PCR with primers MG1655_fw and MG1655_rev to detect prophage integration, as described above. PCR products were resolved on 1% agarose gels, and the intensity of the PCR products representing integrated prophage and non-integrated sites was measured with ImageJ 1.52a (http://imagej.nih.gov/ij).

Prophage integration sites were confirmed by inverse PCR (*Ochman et al., 1988*) as follows: 10 ng of DNA derived from colonies of BW25113 and BW25113Δ*xerC* infected with either EC6098 wild type or EC6098Δ*difC* were cut with restriction enzyme *Hin*dIII (NEB) for 1 hr at 37°C. Reactions were heat inactivated at 80°C for 10 min, and then circularized with T4 DNA Ligase (NEB) overnight at 4°C. Primer VP2_rev, binding the 3'-end of VP2 and facing upstream, and primers Circle_1 and Circle_2, which bind to conserved regions downstream of VP2 and face downstream, were used to amplify circularized ligation products using Phusion polymerase (NEB) under the following PCR conditions described above. PCR products were resolved on 1% agarose gels, and all detected bands were extracted and Sanger-sequenced. Phage integration was confirmed by the presence phage and bacterial sequence in a single read.

## Electron microscopy

Two milliliters of high-titer phage lysate were resuspended in PBS, layered on top of a CsCl step gradient (2 ml each of p1.6 to p1.2 in PBS) and centrifuged in a Beckman Coulter Optima L-100k Ultracentrifuge at 24,000 rpm for four hours. After centrifugation, fractions were collected in 0.5 ml steps, and to determine which fractions contained phage, PCR was performed using primers nocode_fw and VP2_rev using 1 µl of each fraction as template. Fractions containing phage were desalted with an Amicon Ultra-2ml Ultracel-30k filter unit and resuspended in water. For electron microscopy, viral suspensions were pipetted onto carbon-coated grids,

negatively stained with 2% uranyl acetate, and imaged with a Tecnai BioTwin TEM operated at 80kV.

## Acknowledgements

The authors thank the Federal Department of Agriculture for providing *Escherichia spp.* strains and DNA, Bentley A Fane for valuable discussion, Steven Kyle and Zachary A Martinez for assistance in experiments, Dwight Romanovicz for assistance with electron microscopy, and Kim Hammond for figure design. This study was funded by NIH award R35GM118038 to HO. The funders had no role in study design, data collection and analysis, decision to publish, or preparation of the manuscript.'

## Additional information

### Funding

| Funder | Grant reference number | Author |
|---|---|---|
| National Institutes of Health | 26-1612-1250 | Howard Ochman |

The funders had no role in study design, data collection and interpretation, or the decision to submit the work for publication.

### Author contributions

Paul C Kirchberger, Conceptualization, Data curation, Formal analysis, Investigation, Visualization, Methodology; Howard Ochman, Resources, Supervision, Funding acquisition, Validation, Project administration

### Author ORCIDs

Paul C Kirchberger (iD) https://orcid.org/0000-0002-8387-6440
Howard Ochman (iD) https://orcid.org/0000-0003-1688-7059

### Decision letter and Author response

Decision letter https://doi.org/10.7554/eLife.51599.sa1
Author response https://doi.org/10.7554/eLife.51599.sa2

## Additional files

### Supplementary files

- Supplementary file 1. Enterobacteria harboring gokushovirus prophages.
- Supplementary file 2. Enterogokushoviral gene order and conservation within *Gokushovirinae*.
- Supplementary file 3. Reference strains used for host-phylogeny.
- Supplementary file 4. Gokushovirus genomes.
- Supplementary file 5. Primers used in this study.
- Transparent reporting form

### Data availability

The data used in this publication (accession numbers, sequences and alignments) are available in the manuscript, its supplementary files and on datadryad.org: https://doi.org/10.5061/dryad.z8w9ghx7s.

The following dataset was generated:

| Author(s) | Year | Dataset title | Dataset URL | Database and Identifier |
|---|---|---|---|---|
| Kirchberger PC | 2020 | Resurrection of a Global, Metagenomically Defined Gokushovirus | https://doi.org/10.5061/dryad.z8w9ghx7s | Dryad Digital Repository, 10.5061/dryad.z8w9ghx7s |

The following previously published datasets were used:

| Author(s) | Year | Dataset title | Dataset URL | Database and Identifier |
|---|---|---|---|---|
| Moreno-Gallego JL, Chou SP, Di Rienzi SC, Goodrich JK, Spector TD, Bell JT, Youngblut ND, Hewson I, Reyes A, Ley RE | 2019 | The virome in adult monozygotic twins with concordant or discordant gut microbiomes | https://www.ncbi.nlm.nih.gov/bioproject/PRJEB29491 | NCBI BioProject, PRJEB29491 |
| University of Alberta | 2018 | metagenomic sequencing of calf gut content-associated microbiota | https://www.ncbi.nlm.nih.gov/bioproject/PRJNA362629 | NCBI BioProject, PRJNA362629 |
| Kostic AD, Gevers D, Siljander H, Vatanen T, Hyötyläinen T, Hämäläinen AM, Peet A, Tillmann V, Pöhö P, Mattila I, Lähdesmäki H | 2015 | A longitudinal analysis of the developing gut microbiome in infants from Finland, Estonia, and Russian Karelia | https://www.ncbi.nlm.nih.gov/bioproject/PRJNA290380/ | NCBI BioProject, PRJNA290380 |
| Ferretti P, Pasolli E, Tett A, Asnicar F, Gorfer V, Fedi S, Armanini F, Truong DT, Manara S, Zolfo M, Beghini F | 2016 | Mother-infant microbiome vertical transmission | https://www.ncbi.nlm.nih.gov/bioproject/PRJNA352475 | NCBI BioProject, PRJNA352475 |
| Bäckhed F, Roswall J, Peng Y, Feng Q, Jia H, Kovatcheva-Datchary P, Li Y, Xia Y, Xie H, Zhong H, Khan MT | 2015 | Dynamics and Stabilization of the Human Gut Microbiome during the First Year of Life | https://www.ncbi.nlm.nih.gov/bioproject/285453 | NCBI BioProject, PRJEB6456 |
| Stockdale SR, Ryan FJ, McCann A, Dalmasso M, Ross PR, Hill C | 2018 | Viral dark matter in the gut virome of elderly humans | https://www.ncbi.nlm.nih.gov/bioproject/385126 | NCBI BioProject, PRJNA385126 |
| Feng Q, Liang S, Jia H, Stadlmayr A, Tang L, Lan Z, Zhang D, Xia H, Xu X, Jie Z, Su L | 2015 | Gut microbiome development along the colorectal adenoma-carcinoma sequence | https://www.ncbi.nlm.nih.gov/bioproject/277324 | NCBI BioProject, PRJEB7774 |
| Shkoporov A, Clooney AG, Sutton TD, Ryan FJ, Daly KM, Nolan JA, McDonnell SA, Khokhlova EV, Draper LA, Forde A, Guerin E | 2019 | Longitudinal study of the human gut virome | https://www.ncbi.nlm.nih.gov/bioproject/PRJNA545408 | NCBI BioProject, PRJNA545408 |

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
