## [Decision Letter]

**Acceptance summary:**

This study combines bioinformatics and experimental work to examine a group of prevalent, yet not well understood, single-stranded DNA phages known mainly by sequencing surveys. By synthesising viral particles and infecting a model bacterial host the authors reveal that these Gokushoviruses can either integrate into the host chromosome or persist in the cytoplasm, strategies that probably promote survival and explain their global distribution and persistence.

**Decision letter after peer review:**

Thank you for submitting your article "Resurrection of a global, metagenomically mefined microvirus" for consideration by *eLife*. Your article has been reviewed by Karla Kirkegaard as the Senior Editor, a Reviewing Editor, and three reviewers. The following individual involved in review of your submission has agreed to reveal their identity: Mya Breitbart (Reviewer #1).

The reviewers have discussed the reviews with one another and the Reviewing Editor has drafted this decision to help you prepare a revised submission.

Summary:

This is a very good study that has a significant impact on our knowledge of single-stranded DNA phages. The work combines bioinformatics and experimental work to characterize and provide new insight regarding the biology of ssDNA Gokushovirinae phages, a subfamily of the Microviridae phages, that are widespread but have been known primarily from metagenomic sequence surveys. The authors use sequence data to successfully obtain circular phage genomes that are introduced into *Escherichia coli* cells for experimental validation. This work is of interest for microbiologists and those working on metagenomics, phages and overall interested in combining strategies to uncover new biology. Some of the most novel and relevant aspects of the work include:

The combined approach of bioinformatics and experimental validation to obtain a tractable model system of great value for additional studies of these and similar phages.

The description of integration mechanisms dependent on host proteins and palindromic dif-motif integration sites.

The ability of these phages to employ both pseudolysogenic or lytic states as key strategies for survival and global distribution.

The identification of a conserved clade of Gokushovirinae phages, proposed to form a new genus named Enterogokushovirus, that infect Enterobacteria and can explain their reported abundance in gut microbiome samples.

Essential revisions:

The reviewers raised a number of concerns that must be addressed before the paper can be accepted.

There are several questions regarding the nature of the phages obtained and the experimental system used.

1) Are any of the *Enterobacteriaceae* strains naturally containing these prophages available from culture collections? If so, were attempts made to culture these strains and see if lytic phage production occurs in those strains naturally (following the pseudolysogeny model shown in the artificially produced infections)?

2) Did the authors attempt phage induction via chemicals such as Mitomycin C? If so, please include these experiments. Alternatively, the authors could discuss why this is not promising for this group of phages with a different integration/excision mechanism.

3) The authors need to show the absence of other viruses in the propagation strain. The genome of *E. coli*DH5α has 5 prophages, four of which are intact (https://www.ncbi.nlm.nih.gov/pmc/articles/PMC5843723/). Please provide evidence that the plaques and virions shown in Figure 1 panels B,C, D and E are the Enterogokushovirus and not one of the prophages in the strain.

4) The authors need to comment on why there is no phage integration when the BW25113ΔxerD strain is complemented with a functional version of xerD in a plasmid.

5) What is the frequency of integration in the case of BW25113 and EC6098ΔdifCD?

There were also several issues raised regarding data analysis:

6) It was surprising that only 2 prophages similar to this group were identified from screening the human and cattle gut microbiomes. Were viromes examined as well as cellular microbiomes? Looking at the BioProjects searched, it seems like they may have been included but this should be stated explicitly. It would be extremely interesting to know if this group of enterogokushoviruses is found more often in viromes (i.e., free viral particles) as opposed to cellular metagenomes, which might provide insight about their lifestyles and help support the claims in subsection “Recovering prophages from metagenomes”.

7) The argument presented using the prophages and bacteria phylogenies (Figure 2) is not clear and lacks support. The phylogenies need to be improved as no data is presented to support or reject the hypothesis that Enterogokushovirus infect closely related *E. coli* strains.

8) The definition of Enterogokushovirus in based on ANI >50%, conserved gene order and a prophage distribution limited to *Enterobacteriaceae* host, however no data is shown for the conserved gene order or the host of the presented Enterogokushovirus. Quantitative measurements need to be shown for the number of Enterogokushovirus identified in *Enterobacteriaceae*.

9) It was also suggested that a schematic be added (Discussion section) showing the gene order of the enterogokushoviruses compared to representatives from other major groups of Microviridae (including Alpavirinae and other previously described major groups, even if they have not been recognized by the ICTV).

10) While true that having this gokushovirus model system can help in the exploration of gene function and host range determinants (subsection “Phylogenetic analysis of Gokushovirinae”), the data presented here imply that the hypervariable region in VP1 serves as a host range determinant. However, there isn't sufficient evidence for this and there also appear to be other variable regions in the genome.

11) The authors should make available as a supplementary file the gokushovirus alignment used to make Figure 4 (ideally including the metadata on isolation source).

12) One reviewer considers that a reorganization of the Results sections could improve the manuscript. Currently, the section on phylogeny (subsection “Prophages of enterobacteria form a distinct gokushovirus clade”) interrupts the experimental work on *E. coli* (subsection “Mechanisms of enterobacterial gokushoviruses integration into host genomes” and Subsection “Integration into host genomes is not necessary for long-term persistence”). Perhaps moving the phylogeny section, which includes dif motifs, to the end of the results would provide a general outlook on the prevalence of this mechanism.

13) Third paragraph of the Discussion section should say "substitution rate" instead of mutation rate for the result from Minot et al., 2013. The difference is described e.g. in 10.1038/nrg2323 and perhaps two separate statements could be included, one on mutation rates as described in Sanjuan et al., 2010 and one on substitution rates as described in Minot et al., 2013.

---

## [Author Response]

[…] The ability of these phages to employ both pseudolysogenic or lytic states as key strategies for survival and global distribution.The identification of a conserved clade of Gokushovirinae phages, proposed to form a new genus named Enterogokushovirus, that infect Enterobacteria and can explain their reported abundance in gut microbiome samples.

1) During the course of verifying the identity of our phages, as requested by one of the reviewers, we detected co-infection by a widespread cryptic prophage (phi80/Lula). Therefore, we repeated all experiments that possibly involved co-infected cultures or lysates (Figure 5) using fresh stock and *E. coli* strain BW25113Δ*fhuA*, which lacks the receptor for phi80 infection, as host strain for agar-overlay assays for the determination of plaque forming units. These new experiments strengthened our results as follows: (i) Strains with integrated phage produce considerably fewer phage particles than those infected by phage lacking integration motifs (i.e., pseudolysogenic phages); and (ii) In re-doing the time series analysis, pseudolysogenic phages are actually lost at a high rate, which was previously masked by the production of plaques by phi80. We previously discussed pseudolysogeny as an intermediate step between lytic and lysogenic phages, and these new results indicate a trade-off between two strategies—as a pseudolysogen that produces high numbers of phage particles but faces high risk of extinction vs. a lysogen that produces fewer particles but can stably exist in a host. Based on these new findings, we have changed the time-series figure accordingly and added a discussion of the tradeoffs between lysogenic and pseudolysogenic lifestyles. We also note that the previously reported occasional presence of bullseye colonies was most likely the result of coinfection with phi80 – accordingly, we have taken new photographs of the plaques. We have also clarified our method of detecting potentially lysogenized strains – to do so, we picked surviving colonies formed within confluent plaques, which we had previously described as bullseye colonies, which is not an accurate description after all.

2) As suggested, we obtained live cultures of the four strains from which we synthesized gokushovirus genomes. In accordance with our success in producing phage only from MOD1-EC6098 DNA, this strain was the only whose supernatant produced plaques on K12 strains. (See Figure 3—figure supplement 1).

3) As suggested by the reviewers, we changed the structure of the paper to more clearly separate the computational and experimental work. Additionally, Figure 4 (now Figure 2), and its corresponding analysis, have been moved to the beginning of the Results section.

4) We added a supplementary figure (Figure 1—figure supplement 1) that shows the average nucleotide identities of Enterogokushoviruses, and a Supplementary file 2 showing gene orders relative to other Gokushoviruses. This new figure and table offer better support for the genus-level distinction of these phages.

Essential revisions:The reviewers raised a number of concerns that must be addressed before the paper can be accepted.There are several questions regarding the nature of the phages obtained and the experimental system used.1) Are any of the Enterobacteriaceae strains naturally containing these prophages available from culture collections? If so, were attempts made to culture these strains and see if lytic phage production occurs in those strains naturally (following the pseudolysogeny model shown in the artificially produced infections)?

We detected circularized phage in four strains (MOD1-EC2703, EC5150, EC6098 and EC6163) obtained from the FDA. (See new Figure 3—figure supplement 1B). Among these, only supernatant from MOD1-EC6098 produced plaques on BW25113 strains, in accordance with us succeeding only in producing phages from this strain by synthesis and transformation.

2) Did the authors attempt phage induction via chemicals such as Mitomycin C? If so, please include these experiments. Alternatively, the authors could discuss why this is not promising for this group of phages with a different integration/excision mechanism.

The excision/integration of mobile genetic elements via the XerCD system relies on cell division, which is obstructed by mitomycin. As shown in Figure 4, prophage excision occurs continuously without the presence of mitomycin C, and, as such, induction of lysogens appears unnecessary.

3) The authors need to show the absence of other viruses in the propagation strain. The genome of *E. coli*DH5α has 5 prophages, four of which are intact (https://www.ncbi.nlm.nih.gov/pmc/articles/PMC5843723/). Please provide evidence that the plaques and virions shown in Figure 1 panels B,C, D and E are the Enterogokushovirus and not one of the prophages in the strain.

Supernatants of wild type DH5α, either in the presence or absence of mitomycin C, did not result in the formation of plaques on BW25113 (and vice versa, see negative controls in Figure 3—figure supplement 1. Plaques on BW25113 were only observed from supernatants of DH5α that had been transformed with circularized EC6098 (see Figure 3—figure supplement 1A). To ensure purity of phages, each was purified three time by resuspending individual plaques grown on BW25113 in buffer and reinfecting BW25113.

4) The authors need to comment on why there is no phage integration when the BW25113ΔxerD strain is complemented with a functional version of xerD in a plasmid.

Despite multiple attempts, we were unable to restore phage integration using the original XerD plasmid and new constructs. This result is now included in the Discussion section.

5) What is the frequency of integration in the case of BW25113 and EC6098ΔdifCD?

This is our mistake… there is no integration. The mutant was labelled erroneously as EC6098Δdif in Table 1, and we have corrected this error.

There were also several issues raised regarding data analysis:6) It was surprising that only 2 prophages similar to this group were identified from screening the human and cattle gut microbiomes. Were viromes examined as well as cellular microbiomes? Looking at the BioProjects searched, it seems like they may have been included but this should be stated explicitly. It would be extremely interesting to know if this group of enterogokushoviruses is found more often in viromes (i.e., free viral particles) as opposed to cellular metagenomes, which might provide insight about their lifestyles and help support the claims in subsection “Recovering prophages from metagenomes”.

We now describe the nature of the investigated metagenomes (cellular vs enriched for viruses) and have added an additional from recent human virome study (Shkoporov et al., 2019). However, we were not able to assemble full enterogokushovirus MAGs from this dataset. We are currently investigating more metagenomes, but this is beyond the scope of this publication.

7) The argument presented using the prophages and bacteria phylogenies (Figure 2) is not clear and lacks support. The phylogenies need to be improved as no data is presented to support or reject the hypothesis that Enterogokushovirus infect closely related *E. coli* strains.

We have revamped the entire paragraph to make it clearer.

8) The definition of Enterogokushovirus in based on ANI >50%, conserved gene order and a prophage distribution limited to Enterobacteriaceae host, however no data is shown for the conserved gene order or the host of the presented Enterogokushovirus. Quantitative measurements need to be shown for the number of Enterogokushovirus identified in Enterobacteriaceae.

We have added Figure 1—figure supplement 1, which shows the average pairwise nucleotide identities of enterogokushoviruses. Enterogokushovirus hosts are presented in Figure 1 and Supplementary file 1.

9) It was also suggested that a schematic be added (Discussion section) showing the gene order of the enterogokushoviruses compared to representatives from other major groups of Microviridae (including Alpavirinae and other previously described major groups, even if they have not been recognized by the ICTV).

We had added a Supplementary file 2 showing gene order of Enterogokushoviruses and representatives of other gokushoviruses and other microviruses

10) While true that having this gokushovirus model system can help in the exploration of gene function and host range determinants (subsection “Phylogenetic analysis of Gokushovirinae”), the data presented here imply that the hypervariable region in VP1 serves as a host range determinant. However, there isn't sufficient evidence for this and there also appear to be other variable regions in the genome.

This portion of the Discussion section has been edited and removed.

11) The authors should make available as a supplementary file the gokushovirus alignment used to make Figure 4 (ideally including the metadata on isolation source).

We have uploaded all trees and alignments to datadryad.org, https://doi.org/10.5061/dryad.z8w9ghx7s

12) One reviewer considers that a reorganization of the Results sections could improve the manuscript. Currently, the section on phylogeny (subsection “Prophages of enterobacteria form a distinct gokushovirus clade”) interrupts the experimental work on *E. coli* (subsection “Mechanisms of enterobacterial gokushoviruses integration into host genomes” and Subsection “Integration into host genomes is not necessary for long-term persistence”). Perhaps moving the phylogeny section, which includes dif motifs, to the end of the results would provide a general outlook on the prevalence of this mechanism.

We reorganized the manuscript as suggested.

13) Third paragraph of the Discussion section should say "substitution rate" instead of mutation rate for the result from Minot et al., 2013. The difference is described e.g. in 10.1038/nrg2323 and perhaps two separate statements could be included, one on mutation rates as described in Sanjuan et al., 2010 and one on substitution rates as described in Minot et al., 2013.

The reviewers are correct and the sentence has been revised.